# Subcutaneous and Visceral Adipose-Derived Mesenchymal Stem Cells: Commonality and Diversity

**DOI:** 10.3390/cells8101288

**Published:** 2019-10-21

**Authors:** Andreas Ritter, Alexandra Friemel, Susanne Roth, Nina-Naomi Kreis, Samira Catharina Hoock, Babek Khan Safdar, Kyra Fischer, Charlotte Möllmann, Christine Solbach, Frank Louwen, Juping Yuan

**Affiliations:** Division of Obstetrics and Prenatal Medicine, Department of Gynecology and Obstetrics, University Hospital, Goethe University, D-60590 Frankfurt, Germany; alexandra.friemel@kgu.de (A.F.); susanne.roth@kgu.de (S.R.); nina-naomi.kreis@kgu.de (N.-N.K.); samirahoock@gmx.de (S.C.H.); babek.safdar@hotmail.de (B.K.S.); kyra.fischer@kgu.de (K.F.); charlottejohanna.moellmann@kgu.de (C.M.); christine.solbach@kgu.de (C.S.); frank.louwen@kgu.de (F.L.)

**Keywords:** adipose-derived mesenchymal stem cells, differentiation, migration, secretion, primary cilium, sonic hedgehog signaling

## Abstract

Adipose-derived mesenchymal stem cells (ASCs) are considered to be a useful tool for regenerative medicine, owing to their capabilities in differentiation, self-renewal, and immunomodulation. These cells have become a focus in the clinical setting due to their abundance and easy isolation. However, ASCs from different depots are not well characterized. Here, we analyzed the functional similarities and differences of subcutaneous and visceral ASCs. Subcutaneous ASCs have an extraordinarily directed mode of motility and a highly dynamic focal adhesion turnover, even though they share similar surface markers, whereas visceral ASCs move in an undirected random pattern with more stable focal adhesions. Visceral ASCs have a higher potential to differentiate into adipogenic and osteogenic cells when compared to subcutaneous ASCs. In line with these observations, visceral ASCs demonstrate a more active sonic hedgehog pathway that is linked to a high expression of cilia/differentiation related genes. Moreover, visceral ASCs secrete higher levels of inflammatory cytokines interleukin-6, interleukin-8, and tumor necrosis factor α relative to subcutaneous ASCs. These findings highlight, that both ASC subpopulations share multiple cellular features, but significantly differ in their functions. The functional diversity of ASCs depends on their origin, cellular context and surrounding microenvironment within adipose tissues. The data provide important insight into the biology of ASCs, which might be useful in choosing the adequate ASC subpopulation for regenerative therapies.

## 1. Introduction

For several decades, adipose tissue (AT) was thought to be a passive organ with functions in energy homeostasis, accumulation of lipids as energy-storage depot, and supplying energy-rich fat molecules for generating energy and membrane synthesis [1,2]. Nowadays, AT is known to be an important endocrine organ with diverse functions in multiple cellular processes. Cytokine and hormone signals convey immune functions and inflammatory responses that module the energy homeostasis by regulating the food intake, insulin sensitivity, and energy expenditure in tight association with other organs [3]. In line with these complex roles, the ATs of different depots display heterogeneity in their morphological, molecular, and metabolic profiles tightly adjusted to their biological context [4]. Visceral AT in the mesentery and omentum contains increased the numbers of inflammatory and immune cells, is more metabolically active, has an higher uptake of free fatty acids and glucoses, and is less sensitive to insulin [5]. By contrast, subcutaneous AT has a higher affinity for free fatty acids and triglycerides, and its primary function is the energy storage, protection against mechanical damage, and homeostatic heat control [6].

Adipocytes within AT originate from the differentiation process of multipotent progenitor cells, named adipose-derived mesenchymal stem cells (ASCs) [7]. These cells are key regulators of AT. They are involved in tissue homeostasis with their potent differentiation capacity in adipogenesis and angiogenesis. Additionally, ASCs coordinate and maintain the local and systemic environment by immunomodulation and damage repair through their paracrine signaling and direct cell-cell interaction [8,9]. These cells are considered to be useful for novel regenerative medicine applications due to their accessibility and their various functions concerning tissue remodeling and homeostasis and become the focus of many translational clinical studies [10].

Numerous studies have been conducted to analyze the key properties of ASCs isolated from different AT depots (visceral, subcutaneous, and preperitoneal) [11]. In particular, subcutaneous and visceral ASCs have received great attention. However, the results of these studies were inconsistent regarding their differentiation potential, proliferation, and paracrine signaling [11,12]. We have already investigated the interaction of ASCs with breast cancer cells [13], the effect of Polo-like kinase 1 (Plk1) inhibitors on ASCs [14], and the influence of obesity on ASCs [15,16]. During these studies, we have observed similarities and disparities between subcutaneous and visceral ASCs. In the present work, we focus on the features of both ASC subtypes and further analyze their main functions. Our results highlight similarities in their cell surface marker profile, cell viability, and cell cycle progression, and diversities in the motility, differentiation capacity, and the cilium related sonic hedgehog (Hh) signal pathway.

## 2. Materials and Methods

### 2.1. Human Visceral and Subcutaneous ASC Isolation and Cell Surface Marker Measurement

This work was approved by the Ethics Committee of the Johann Wolfgang Goethe University Hospital Frankfurt and informed written consent was obtained from all the participants. Visceral (omental) and subcutaneous (abdominal) adipose tissues were taken from women undergoing Caesarean section. Table 1 lists participant information. Their age ranged between 25 and 35 years. Body mass index (BMI) of both ASC subgroups was 24.1 ± 2.9. ASCs were isolated, as described previously [13,14]. After isolation, the cells were cultured and expanded for three passages. Cells were then stored at −80 °C until use. Early passages (P3 to P6) of ASCs were used for all analyses. All experiments, unless otherwise indicated, were independently performed with ASCs that were isolated from at least three different donors.

FACSCalibur^TM^ (BD Biosciences, Heidelberg, Germany) was used for determining the surface markers of ASCs. The cells were harvested with 0.25% trypsin and fixed for 15 min. with ice-cold 2% paraformaldehyde (PFA) at 4°C. Cells were washed twice with flow cytometry buffer (FCB: PBS with 0.2% Tween-20, and 2% fetal calf serum (FCS)) and stained with the following antibodies from eBioscience/BD-Pharmingen (Frankfurt, Germany): FITC-conjugated anti-human cluster of differentiation 90 (CD90) (#11-0909-42), PE-conjugated anti-human CD73 (#550257), PE-conjugated anti-human CD 105 (#323206), PE-conjugated anti-human CD146 (#561013), PerCP-Cy5.5-conjugated anti-human CD14 (#555397), FITC-conjugated anti-human CD34 (#343504), APC-conjugated anti-human CD106 (#551147), and APC-conjugated anti-human CD31 (#17-0319-41). Anti-mouse Ig, κ/negative control compensation particles (eBioscience/BD-Pharmingen, #552843), flow cytometry setup beads (eBioscience/BD-Pharmingen, #340486 and #340487), and non-stained ASCs were used as negative controls for FACS gating.

### 2.2. Indirect Immunofluorescence Staining, Microscopy, Fluorescence Intensity Quantification, and Nocodazole Washout

Indirect immunofluorescence staining was performed, as reported [17]. Cells were seeded on Nunc^TM^ Lab-Tek^TM^ SlideFlask chambers from Thermo Fisher Scientific (Schwerte, Germany). Cells were fixed for 8–10 min. with methanol at −20 °C or with 4% PFA containing 0.2% Triton X-100 for 15 min. at room temperature, as described [16]. The following primary antibodies were used: mouse monoclonal antibody against acetylated α-tubulin (Sigma-Aldrich, Darmstadt, Germany, #T6793), mouse monoclonal antibody against CD90 (Abcam, Cambridge, UK, ab133350), rabbit monoclonal antibody against CD73 (GenTex, Eching, Germany, GTX101140), mouse monoclonal antibody against Smo (Santa Cruz Biotechnology, Heidelberg, Germany, #sc-166685), rabbit polyclonal antibody against phospho-histone H3 (pHH3, Ser10, Merck Millipore, Darmstadt, Germany, #06-570), mouse monoclonal antibody against phospho-focal adhesion kinase (p-FAK, Cell Signaling, Frankfurt, Germany, #3283), rabbit monoclonal antibody against FAK (Proteintech, Manchester, UK, 66258-1-Ig), mouse monoclonal antibody against paxillin (BD Transduction Laboratories^TM^, Frankfurt, Germany, #610619), and rabbit polyclonal antibody against p-paxillin (Cell Signaling, Frankfurt, Germany, #2541). FITC-, Cy3-, and Cy5 conjugated secondary antibodies were obtained from Jackson ImmunoResearch (Cambridgeshire, UK). DNA was visualized by using DAPI (4′,6-diamidino-2-phenylindole-dihydrochloride, Roche, Mannheim, Germany). The filamentous actin (F-actin) cytoskeleton was stained while using phalloidin (Phalloidin-Atto 550; Sigma-Aldrich, Munich, Germany). The slides were examined while using an AxioObserver.Z1 microscope (Zeiss, Göttingen, Germany) and images were taken using an AxioCam MRm camera (Zeiss, Göttingen, Germany). The immunofluorescence stained slides were further examined by confocal laser scanning microscopy (CLSM) using Z-stack images with a HCXPI APO CS 63.0 x 1.4 oil objective (Leica CTR 6500, Heidelberg, Germany) in sequential excitation of fluorophores. A series of Z-stack images were captured at 0.5 µm intervals. All the images in each experiment were taken with the same laser intensity and exposure time. Representatives are generated by superimposing (overlay) individual images from confocal Z-sections.

Fluorescent intensity was measured while using line-scan-based analysis in ImageJ (National Institutes of Health), as described [15,18]. The average intensities over a three-pixel-wide line along the axoneme were measured and then normalized against cilium length by using the ImageJ plugin Plot Roi Profile. The intensity was measured from the axonemal base to its tip in 10% intervals. The mean values of thirty cilia from three different donors were obtained for each group within the intervals and were plotted to GraphPad Prism 7 (GraphPad software Inc., San Diego, USA).

We performed a nocodazole washout assay to analyze the dynamics (disassembly/reassembly) of focal adhesions (FAs). The cells were treated with nocodazole (10 µM; Sigma-Aldrich, Darmstadt, Germany) for 5 h to depolymerize microtubules (MTs) [19]. The drug was washed out with phosphate-buffered saline (PBS), and MTs were repolymerized in medium for different time periods (0, 30, 75 min.). Cells were fixed and stained for paxillin and p-FAK. The slides were examined while using an AxioObserver.Z1 microscope (Zeiss, Göttingen, Germany) and the images were taken using an AxioCam MRm camera (Zeiss, Göttingen, Germany).

### 2.3. Sonic Hedgehog Stimulation, Cytokine Array and ELISA

Cells were incubated with 200 nM smoothened agonist (SAG) (Bioscience, Wiesbaden, Germany) in the absence of FCS for 24 h for activating the Hedghog (Hh) pathway. Immunofluorescence line-scan-based analysis and quantitative RT-PCR analysis were then performed [15].

For cytokine measurement, visceral and subcutaneous ASC in passage 3 were cultured for three days to a confluence of 90% and supernatants were taken. The levels of chemokines, cytokines, and growth factors in the supernatants were determined by applying a human cytokine antibody array according to the manufacturer’s instructions (R&D, Wiesbaden, Germany). The chemiluminescent membranes were developed using the ChemiDoc^TM^ MP System (Bio-Rad, Munich, Germany) and the signal intensity was assessed with ImageJ 1.49i software (National Institutes of Health, Bethesda, USA) by determining the pixel intensity of the detected spots. The signal value from the provided negative control was subtracted from every measured sample [13].

The 72 h supernatants of visceral or subcutaneous ASCs were also used for evaluating IL-6, IL-8, and TNFα via ELISA, as instructed by the manufacturers (PeproTech, Hamburg, Germany).

### 2.4. Cell Cycle Analysis and Cell Proliferation

The cell cycle distribution was analyzed using a FACSCalibur^TM^ (BD Biosciences, Heidelberg, Germany), as reported [20]. Briefly, cells were harvested, washed with PBS, fixed in chilled 70% ethanol at 4 °C for 30 min., treated with 1 mg/mL of RNase A (Sigma-Aldrich, Munich, Germany) and stained with 100 μg/mL of propidium iodide (PI) for 30 min. at 37 °C. DNA content was determined.

Cell proliferation assays were carried out by using Cell Titer-Blue^®^ Cell Viability Assay (BD Biosciences, Heidelberg, Germany) on treated cells in 96-well plates (Promega, Mannheim, Germany). 20 μL of CellTiter-Blue^®^ reagent was added to each well and then incubated at 37 °C with 5% CO_2_ for 4 h before fluorescence reading while using a Victor 1420 Multilabel Counter (Wallac, Finland), as reported [20].

### 2.5. ASC Differentiation And Western Blot Analysis

ASC differentiation was performed, as reported [13]. ASCs were cultured with StemMACS AdipoDiff media (Miltenyi Biotec, Gladbach, Germany) up to 14 days to induce adipogenic differentiation. Cells were then fixed and stained for adiponectin (Abcam, Cambridge, #ab22554) and analyzed for lipid droplets characteristic of adipocytes. For osteogenic differentiation, ASCs were incubated with StemMACS OsteoDiff media (Miltenyi Biotec, Gladbach, Germany) up to 14 days, fixed, and stained with 2% Alizarin Red S (pH 4.2) to visualize calcific deposition, a hallmark of osteogenic cells.

Western blot analysis was performed, as reported [21,22], with the following antibodies: mouse monoclonal antibodies against cyclin B1 (Santa Cruz Biotechnology, Heidelberg, Germany, GNS1), monoclonal mouse against p53 (Santa Cruz Biotechnology, Heidelberg, Germany, DO-8), and rabbit polyclonal antibodies against cyclin B1 (Santa Cruz Biotechnology, Heidelberg, Germany, H-433), mouse monoclonal antibodies against p21 (Cell Signaling, Frankfurt, Germany, DSC60), rabbit monoclonal antibodies against E-cadherin (Cell Signaling, Frankfurt, Germany, 24E10), mouse monoclonal antibodies against vimentin (Dako, Hamburg, Germany, M0725), rabbit monoclonal antibody against fibronectin (BD Biosciences, 1573-1), mouse monoclonal β-actin (A2228) (Sigma-Aldrich, Munich, Germany), and GAPDH (GTX627408) from GeneTex (Eching, Germany).

### 2.6. RNA Extraction And Real-Time PCR

Total RNAs of ASCs were extracted with RNeasy Mini kit (7Bioscience, Neuenburg, Germany). Reverse transcription was performed while using High-Capacity cDNA Reverse Transcription Kit (Promega, Mannheim, Germany), as instructed. All the probes for gene analysis were obtained from Applied Biosystems: ADIPOQ (#Hs00605917_m1), IL-6 (#Hs00985639_m1), IL-8 (#00175123_m1), IL-10 (#Hs00961622_m1), TNFα (Hs00174128_m1), PLK1 (#Hs00153444_m1), PLK4 (#Hs00179514_m1), KIF2A (#Hs00189636_m1), SMO (#Hs01090242_m1), GLI1 (#Hs00171790_m1), NANOG (#Hs04260366_g1), PTCH1 (#Hs00181117_m1), RUNX2 (#Hs01047973_m1), KLF4 (#Hs00358836_m1), c-MYC (#Hs00153408_m1), PPARγ (#Hs01115513_m1), LEPTIN (#Hs00174877_m1), SOX2 (#Hs01053049_s1), EpCAM (Hs0090188s_m1), VIM (#Hs00958111_m1), SNAIL1 (Hs00195591_m1), TWIST (Hs01675818_s1), ZEB1 (Hs01566408_m1), and GAPDH (#Hs02758991_g1). Real-time PCR was performed with a StepOnePlus Real-time PCR System (Applied Biosystems). The data were analyzed while using StepOne Software v.2.3 (Applied Biosystems), as described previously [23].

### 2.7. Cell Motility, Migration, Attraction, and Invasion

For motility assay, the cells were seeded into 24-well plates with a low confluency and they were imaged for 12 h at 5 min. time intervals. All time-lapse imaging was performed with an AxioObserver.Z1 microscope (Zeiss, Göttingen, Germany) and imaged with an AxioCam MRc camera (Zeiss, Göttingen, Germany) that was equipped with an environmental chamber to maintain proper environmental conditions (37 °C, 5% CO_2_). The time-lapse movies were analyzed by using ImageJ 1.49i software (National Institutes of Health) with the manual tracking plugin, and Chemotaxis and Migration Tool (Ibidi GmBH, Munich, Germany). The tracks were derived from raw data points and they were plotted in GraphPad Prism 7 (GraphPad software Inc.). The accumulated distance was calculated by using the raw data points by the Chemotaxis and Migration Tool. Thirty random cells per experiment were analyzed and the experiments were independently repeated three times. The patterns of motility were evaluated, as described previously [15,16,24].

Cell migration assays were performed with culture-inserts from ibidi (Martinsried, Germany). Visceral or subcutaneous ASCs (6.5 × 10^4^) cells were seeded in each well of the culture-inserts. Culture-inserts were gently removed after at least 8 h. The cells were acquired and imaged at indicated time points with bright-field images. Four pictures of each insert were taken (three inserts for each experimental condition) and the experiments were performed in triplicates. The open area was measured while using the AxioVision SE64 Re. 4.9 software (Zeiss, Göttingen, Germany).

For attraction assay, cells were placed in six-well plates and one well of each insert was filled with visceral or subcutaneous ASCs (5.5 × 10^4^) or with the investigated cells (MCF-7 or MDA-MB-231). After 8 h, the culture-inserts were removed and the images were obtained at the indicated time points. Cellular movement toward other migration front was evaluated by measuring the distance between the cell nucleus and the outermost cellular protrusion using the AxioVision SE64 Re. 4.9 software (Zeiss, Göttingen, Germany). The experiments were independently performed three times.

Visceral or subcutaneous ASCs were seeded (7.5 × 10^4^) in 24-well transwell matrigel chambers for invasion assay, according to the manufacturer’s instructions (Cell Biolabs Inc, San Diego), as previously reported [13]. The cells were fixed with ethanol and stained with DAPI. The invaded cells were counted with a microscope. The experiments were independently performed three times.

### 2.8. Statistical Analysis

Student’s t-test (two tailed and paired or homoscedastic) was used to evaluate the significance of difference between different groups for gene analysis, cell viability assay, cell cycle distribution, and ciliated cell population. An unpaired Mann–Whitney *U* test was used to perform the statistical evaluation of the single cell tracking assay, line-scan analysis, and the measurement of the cilium length (two tailed). Difference was considered to be statistically significant when *p* < 0.05.

## 3. Results

### 3.1. Subcutaneous and Visceral ASCs Have a Comparable Cell Surface Marker Profile and Proliferation Rate

For this study, subcutaneous and visceral ASCs were isolated from age and BMI matched donors undergoing Caesarean sections. Table 1 lists the clinical information.

The expression of cell surface markers was compared between subcutaneous ASCs (ASCsub) and visceral ASCs (ASCvis). The indirect immunofluorescence staining of CD90 and CD73, two important markers for mesenchymal stem cells (MSCs), including ASCs, showed positive signals to a comparable extent in both types of ASCs (Figure 1A). This was further underscored by flow cytometric analyses of CD90, CD73, CD146, and CD105 as positive markers and CD14, CD31, CD106, and CD34 as negative markers (Figure 1B), as described [25,26]. The percentages of cell surface markers were comparable between ASCsub and ASCvis (Figure 1B). Additionally, the cell cycle distribution of both ASC subtypes differed by only 3% in G0/G1-phase (ASCsub: 72%, ASCvis: 69%) and G2/M-phase (ASCsub: 18%, ASCvis: 15%) (Figure 1C,D). Furthermore, the cells were stained for phospho-histone H3 (pHH3 (Ser10)) (Figure 1E), a mitotic marker, for the evaluation of mitotic cells. No significant difference in the mitotic cell population was observed between two subtypes of ASCs (Figure 1F). ASCs were also harvested for Western blot analysis. The important mitotic proteins cyclin B1 and Aurora A showed no differences in their protein expression (Figure 1G, lane 1 and 2), whereas the cellular stress response proteins p53 and p21 were slightly elevated in visceral ASCs (Figure 1G, lane 4 and 5). Finally, the subcutaneous ASCs showed marginally increased cell viability upon 72 h and 96 h, which could not reach a significant level (Figure 1H, 72 h and 96 h). In summary, the results reveal no significant differences between matched ASCsub and ASCvis cells in the expression of their cell surface markers, cell cycle distribution, important mitotic regulators, and cell viability.

### 3.2. Both ASC Subtypes Display Comparable Migration and Invasion Rates but Different Modes of Motility

An important function of MSCs, to which ASCs belong, is their migration and homing ability to a target tissue or cell type [27]. Several assays were performed with ASCsub and ASCvis to investigate this issue in detail. At first, wound healing/migration assays were carried out, as previously described [13]. After 16 and 24 h, ASCsub cells showed a significantly increased migration capacity when compared to ASCvis (Figure 2A,B). Surprisingly, this pattern could not be observed in a single-cell tracking experiment, which analyzes the random movement of cells up to 12 h, as reported [14]. The results illustrated even a slightly increased accumulated distance in visceral ASCs with 523 µm as compared to 449 µm in subcutaneous ASCs (Figure 2C, left graph, and D). As calculated by time and distance, the velocity of these cells was also moderately elevated (ASCvis: 0.78 µm/min., ASCsub: 0.67 µm/min.) (Figure 2C, middle graph). In contrast, the intrinsic directionality of ASCvis was significantly reduced when compared to ASCsub (ASCvis: 0.23 d/D, ASCsub: 0.62 d/D) (Figure 2C, right graph), indicating that the ASCsub cells move in a directed fashion. To study the motility in a three-dimensional (3D) environment, invasion assays were carried out [13]. Interestingly, 39.6% of ASCsub and 40.3% of ASCvis were able to invade through the matrigel layer (Figure 2E,F) and no significant difference (*p* = 0.49) could be observed between both subpopulations.

Many studies have reported a tropism of ASCs toward cancer cells [28], which can be demonstrated via homing experiments [13]. Homing experiments were performed with ASCs and breast cancer cells seeded in separated chambers of a culture insert with a defined cell free gap to corroborate the observation that ASCsub cells move in a more directed fashion compared to ASCvis. We measured the cell homing distance, the length between the nucleus, and the outermost cell protrusion, including lamellipodia and filopodia, of ASCs toward distinct breast cancer cell lines MCF-7 and MDA-MB-231 demonstrating the tropism of ASCs at 8 h and 16 h. ASCsub showed a significantly increased homing ability after 8 h and 15 h toward MCF-7 and MDA-MB-231 cell lines as compared to ASCvis (Figure 2G, left and middle graphs). Interestingly, subcutaneous ASCs home even to their own cell type (Figure 2G,H, right graphs). In contrast, visceral ASCs were homing less efficiently toward other cell types (Figure 2G,H). Collectively, these results strongly suggest that subcutaneous ASCs have an extraordinarily directed mode of motility, whereas visceral ASCs seem to move in an undirected random fashion.

### 3.3. Subcutaneous ASCs Display Higher Gene Levels of Mesenchymal Transcription Factors and Have More Dynamic and Composition Altered FAs Compared to Visceral ASCs

Both subtypes of ASCs displayed distinct morphologies that are shown by staining with phalloidin, an actin cytoskeleton dye, and paxillin, a focal adhesion marker, especially in their early 3–4 passages. Subcutaneous ASCs were more “classical” mesenchymal-like, showing characteristics of a fibroblast-like morphology with a small and long cell body (Figure 3A, upper panel). Visceral ASCs were less mesenchymal-like with some likeness of apical-basal polarity and a large cell body (Figure 3A, lower panel). Western blot analyses corroborated this notion, by showing less protein levels of fibronectin and vimentin (Figure 3B, lane 1 and 3), mesenchymal cell markers, and higher levels of E-cadherin (Figure 3B, lane 2), an epithelial cell marker, in visceral ASCs as compared to subcutaneous ASCs (Figure 3B). In line with these data, the gene levels of zinc finger E-box-binding homeobox 1 (*ZEB1)*, snail family transcriptional repressor 1 (*SNAIL1),* and twist family basic helix-loop-helix (bHLH) transcription factor 1 (*TWIST)*, three important mesenchymal transcription factors, were enriched in ASCsub when compared to ASCvis (Figure 3C). Moreover, the epithelial cell adhesion molecule (*EpCAM*) gene was more demonstrative in ASCvis, while the expression of mesenchymal gene vimentin (*VIM*) encoding a type III intermediate filament was higher in ASCsub (Figure 3D, left) (Figure 3D, right). These results suggest that subcutaneous and visceral ASCs differ in the expression of mesenchymal and epithelial markers contributing to the different migration behavior of these cells (Figure 2 and Figure 3A–D).

The FAs of both ASC types were analyzed in depth by immunofluorescence staining of focal adhesion kinase (FAK), phospho-FAK (p-FAK), paxillin, and p-paxillin (Figure 3E–K), the most important components of FAs, since the protein expression, phosphorylation status, and the FA composition are influenced by a variety of transcription factors, including *TWIST*, *SNAIL1,* and *ZEB1* [29,30]. Though showing no difference in p-paxillin signal, subcutaneous ASCs displayed significantly less mean fluorescence intensity of FAK (88 a.u.), p-FAK (123 a.u.), and paxillin (60 a.u.), in comparison with visceral ASCs (FAK (123 a.u.), p-FAK (131 a.u.), and paxillin (68 a.u.)) (Figure 3E–K). Furthermore, the size of FA correlates with its stability [31,32]. Indeed, the FA size was significantly enlarged in ASCvis (ASCvis; 7.07 µm), relative to ASCsub (5.66 µm) (Figure 3I).

More FA components (Figure 3E–H) and enlarged FA size (Figure 3I) indicate altered FA dynamics of visceral ASCs. We performed a nocodazole washout experiment to delineate this issue, in which MT depolymerization induced by nocodazole leads to stabilized FAs through stress fiber formation and the release from the nocodazole treatment triggers FA disassembly through rapid MT regrowth [33]. ASCs were subjected to nocodazole (10 µM), released for 0, 30, and 75 min., stained for p-FAK and paxillin, two important FA assembly factors, for microscopic evaluation (Figure 4A). Subcutaneous ASCs dynamically responded to the nocodazole release evidenced by rapidly decreased paxillin, p-FAK, and FA size at 30 min. Followed by efficiently recruiting these FA components, leading to enlarging FA size at 75 min. (Figure 4B–E, left panels). In line with the observations (Figure 3F,H,I), visceral ASCs showed high levels of paxillin and p-FAK and the large FAs, even after the cells were treated with nocodazole (Figure 4B–E, right panels, 0 h). Furthermore, despite disassembled FAs being triggered by the nocodazole release at 30 min., visceral ASCs had difficulty in reassembling their FAs at 75 min. by showing low p-FAK level and small FA size (Figure 4B–E, right panels). In sum, these results clearly suggest that visceral ASCs have stabilized FAs with a static turnover, whereas subcutaneous ASCs confer dynamic FAs, which contributes to their efficiently directed motility.

### 3.4. Visceral ASCs Have a Higher Osteogenic and Adipogenic Differentiation Capacity and Upregulated Levels of Stemness-Like Associated Genes

The differentiation capacity of ASCs that were isolated from different adipose tissue depots is the subject of controversial debates over the last decade, with multiple investigations reporting inconsistent results [5,6,11]. This could be ascribed to the usage of ASCs that were isolated from distinct donors with varied age, BMI, and gender, different isolation methods or varying ASC passages. We used matched ASCs from the same donor and in identical passage for differentiation experiments. We first analyzed multiple classical stemness/self-renewal associated genes, like myelocytomatosis proto-oncogene cellular homolog (*c-MYC)*, SRY-box transcription factor 2 (*SOX2)*, Krüppel-like factor 4 (*KLF4),* and *NANOG*, which are also known for their crucial role in osteogenic and adipogenic differentiation of ASCs [34,35,36]. In fact, ASCvis showed a significantly higher expression of stemness associated genes, like *c-MYC, SOX2*, *KLF4,* and *NANOG,* when compared to ASCsub (Figure 5A).

ASCs were then differentiated to osteocytes or adipocytes for up to 14 days. The cells were stained with Alizarin Red S to visualize calcific deposition (osteogenic lineage) or adiponectin (adipogenic lineage). Relative to ASCvis (Figure 5C,F, lower panels), ASCsub showed lower numbers of positive cells that were stained with Alizarin Red S (Figure 5C, upper panel) or adiponectin (Figure 5F, upper panel). Further analysis revealed that, as compared to visceral ASCs with 24.4% osteogenic-like and 45.5% adipogenic-like cells, subcutaneous ASCs displayed 15.7% osteogenic-like and 35.6% adipogenic-like cells (Figure 5B,E). Total RNAs from these cells were also isolated for gene analysis. ASCvis showed enhanced levels of osteogenic related genes, including protein patched homolog 1 (*PTCH1)* and runt-related transcription factor (*RUNX2)* (Figure 5D, 1^st^ and 2^nd^ graph), as well as *c-MYC* (Figure 5D, 3^rd^ graph). Interestingly, *KLF4*, which is a negative regulator for osteogenic differentiation, was upregulated in subcutaneous ASCs after differentiation (Figure 5D, 4^th^ graph). Similar results were also obtained for adipogenic differentiation. Adiponectin (*ADIPOQ*), *LEPTIN,* and peroxisome proliferator activated receptor gamma (*PPARγ*), three adipogenic related genes, were increased in ASCvis upon differentiation (Figure 5G). These data strengthen the notion that visceral ASCs have an improved differentiation capability that is likely associated with a higher gene expression of stemness associated genes.

### 3.5. Subcutaneous ASCs are More Ciliated but Have A Less Active Sonic Hedgehog (Hh) Pathway

The primary cilium is a sensor organelle that is critical for responding to a variety of extra and intracellular stimuli as a central processing unit [37]. It is connected to many key functions of ASCs, like migration, secretion, and differentiation [38,39], and indispensable for the Hh signaling in mammals [40]. To compare the cilium size as well as the ciliated cell population between ASCsub and ASCvis, the cells were stained for acetylated α-tubulin, adenosine diphosphate ribosylation factor-like GTPase 13B (Arl13b), two typical cilium markers, and DNA followed by microscopic analysis (Figure 6A). The primary cilium size was comparable between both ASC subtypes with a mean cilium length of 4.61 µm in ASCsub and 4.60 µm in ASCvis (Figure 6A,B). While 28.6% of visceral ASCs were ciliated, the primary cilium was present in 43.9% of subcutaneous ASCs (Figure 6C). In line with less ciliated cells, the gene levels of Polo-like kinase 1 (*PLK1)*, Polo-like kinase 4 (*PLK4),* and kinesin family member 2A (*KIF2A)*, genes that are responsible for deciliation and here independent from their roles in mitosis, are upregulated in visceral ASCs (Figure 6D).

The Hh signaling pathway is exclusively mediated by the primary cilium and it is involved in the differentiation processes of multiple types of various stem/mesenchymal stem cells, including ASCs [38,41]. The activation of the Hh pathway is a signaling cascade recruiting Smoothened (Smo) and glioma-associated oncogene homolog 1–3 (Gli1-3) to the primary cilium, which leads to an accumulation of these proteins on the proximal base and distal tip of the cilium [42]. ASCs were treated with 200 nM of SAG, a smoothened agonist, for 16 h, to compare the activation and response of this pathway between both subtypes. The treated ASCs were stained for Arl13b, Smo, and pericentrin for microscopy. Line scan analysis of fluorescent Smo was performed from the proximal base to the distal tip of cilia (Figure 6F). Subcutaneous ASCs demonstrated a significantly lowered intensity of Smo on the proximal as well as the distal part of the primary cilium when compared to visceral ASCs (Figure 6E), which suggests the hampered recruitment of Smo to the cilium in ASCsub. Furthermore, gene analysis revealed a significantly lower expression of Hh related genes *GLI1*, *PTCH1*, *SMO,* and two downstream targets *TP53* and *NANOG* after 24 h SAG stimulation in subcutaneous ASCs relative to visceral ASCs (Figure 6G). The less active Hh signaling could be an additional explanation for the reduced adipogenic/osteogenic differentiation capacity of subcutaneous ASCs.

### 3.6. Visceral ASCs Secrete More Inflammatory Cytokines

MSCs, especially ASCs, are well known as a source of many secreted cytokines, chemokines, and growth factors regulating diverse cell-cell communications [43]. As reported previously, we analyzed the secretion of multiple factors from ASCsub and ASCvis by using a cytokine array demonstrating the secretion of various inflammation cytokines [13]. To corroborate these results, the concentrations of interleukin 6 (IL-6), interleukin 8 (IL-8), and tumor necrosis factor α (TNFα) were measured in the supernatant of both ASC subtypes by enzyme-linked immunosorbent assay (ELISA). Subcutaneous ASCs secreted significantly less of all three inflammatory cytokines, with 186 pg/mL of IL-6, 676 pg/mL of IL-8, and 111 pg/mL of TNFα when compared to 236 pg/mL of IL-6, 761 pg/mL of IL-8, and 194 pg/mL of TNFα secreted by ASCvis (Figure 7A). In support of these observations, the gene levels of *IL-6*, *IL-8*, *IL-10,* and *TNFα* were higher in visceral ASCs as compared to subcutaneous ASCs (Figure 7B).

## 4. Discussion

ASCs have gained high attention as a promising tool for novel cell-based therapies in the field of regenerative medicine, supported by multiple encouraging preclinical studies in a variety of human diseases [44,45]. However, the diversity among ASCs that were isolated from different adipose tissue depots affects their biological functions and features [5,6,11]. In this work, we have isolated ASCs from subcutaneous and visceral adipose tissue of the same donors and systematically characterized multiple features of these paired ASCs in detail.

The results show that both subcutaneous and visceral ASCs display relatively comparable percentages of cells positive for cell surface markers, which are characteristic for MSCs [25]. Cell viability, cell cycle distribution, and mitotic cell population are also similar between subcutaneous and visceral ASCs, which is in agreement with the results from several groups [46,47,48] and yet inconsistent with the data from other studies [11,49,50,51]. This discrepancy generally results from non-matched donors and varied methodology for ASC isolation/culture. It is observed in many studies that subcutaneous and visceral ASCs were derived from donors without further description of related clinical information, like age, gender, and BMI, or from other species [46,47,48,49,50,52]. These heterogeneous data underscore the requirement of matched donor collectives and standardized protocols. Although it is reported that the proliferation rate of ASCs is hardly changed by the reproductive status [53], we do not exclude the possibility that pregnancy related hormones influence some of ASC features.

One key feature of mesenchymal stem cells is their homing ability toward damaged tissue and to serve in these areas as a reservoir for growth factors and regenerative promoters [54]. We showed that, relative to visceral ASCs, subcutaneous ASCs have a more directed mode of motility, evidenced by single-cell tracking, and especially homing assays, in which ASCs move toward different cell lines, like MCF-7, MDA-MB231, or themselves. Moreover, subcutaneous ASCs display a classical mesenchymal phenotype with a spindle-like form and a small cell body, whereas visceral ASCs show a large cell body and some likeness of an “apical-basal polarity” [55]. In line with these findings, well-known mesenchymal markers, like fibronectin and vimentin, as well as genes of mesenchymal related transcription factors, including *SNAIL*, *SLUG*, *TWIST,* and *ZEB1,* were low in visceral ASCs. This could contribute to the decreased directional movement of visceral ASCs, since an increased mesenchymal phenotype correlates with an enhanced directional motility, as reported for multiple cell lines during the process of the epithelial-to-mesenchymal transition (EMT) [56,57]. The analyses of important FA proteins illustrate a more complex pattern: FAK, p-FAK, and paxillin were significantly higher in visceral ASCs, which is often associated with efficient cellular migration [58]. However, FA size is related to the migration rate and the residence time of FAK and paxillin [31,32]. Indeed, visceral ASCs displayed highly enlarged FAs, as shown in cells switching from an epithelial to mesenchymal phenotype [30], which implies an accumulation of these proteins mediated by a slower FA turnover rate in visceral ASCs. This assumption was further corroborated by the data from the nocodazole washout assay, illustrating more dynamic FAs in subcutaneous ASCs with a higher rate of FA disassembly and reassembly after the nocodazole release as compared to visceral ASCs.

In conclusion, we show that subcutaneous ASCs conduct a more directional movement that is associated with an efficient FA turnover than visceral ASCs, though both subtypes move with comparable rates. This difference could be likely ascribed to their different origins [59] with distinct biological functions. An enhanced homing ability might be important to execute the protective and cell renewal functions of the subcutaneous AT [6], whereas the visceral AT is important in controlling metabolism and inflammatory signals [5], which are less dependent on the homing/migration ability of these cells.

In agreement with this notion, visceral ASCs secreted high levels of different chemokines/cytokines that are involved in autocrine and paracrine regulation. Among these cytokines, IL-6, IL-8, and TNFα were highly secreted by visceral ASCs, which is in line with the results that were derived from a recent report [60]. This substantial secretion of inflammatory cytokines is likely associated with their impact on the surrounding cells and microenvironment, which was previously shown in breast cancer cell lines MCF-7, MDA-MB-231, and non-tumorigenic breast epithelial cells, like MCF-10A [13,61].

Another characteristic property of ASCs is their capacity to differentiate into multiple cell types, including osteocytes and adipocytes [8]. We found that visceral ASCs have an increased differentiation potential toward the osteogenic and adipogenic lineage as compared to subcutaneous ASCs. This could be ascribed to an enhanced gene expression of stemness/self-renewal associated genes *c-MYC*, *KLF4, NANOG,* and *SOX2*, which are reported to be essential for the differentiation of MSCs/ASCs [35,36,62,63]. Especially, *c-MYC* has been shown to be crucial for the initiation of the adipogenic differentiation by regulating key genes, like fatty acid-binding protein 4 (*FABP4*), *PPARγ*, *ADIPOQ,* and *LEPTIN* in ASCs [63]. These results are not consistent with data from other studies, showing that subcutaneous ASCs are more competent in differentiation [50,64], again indicating the inconsistent results in this field. Again, this might be attributable to species differences and non-matched donors. Additionally, we are not able to exclude the impact of donors’ reproductive state, which might enhance this capability of visceral ASCs, although *Ng* et al. have reported no significant differences between isolated ASCs from pregnant, premenopausal, and menopausal donors [53].

Interestingly, an increased number of subcutaneous ASCs were ciliated, which could be associated with their motility, as primary cilia are especially involved in directional migration [65]. However, relative to visceral ASCs, subcutaneous ASCs displayed less activation of the sonic Hh pathway upon stimulation, which is associated with less expression of its downstream targets. The Hh pathway is important for differentiation, since a report showed that the removal of the primary cilia and its associated Hh signaling leads to an increased ASC proliferation and decreased Runx2, alkaline phosphatase, and bone morphogenetic protein-2 mRNA (BMP2) expression, finally reducing their osteogenic differentiation potential [39]. Moreover, we have recently reported that the impaired differentiation capacity of obese ASCs is rescuable through the restoration of the cilium length and the sonic Hh pathway [16]. These data suggest that the less active Hh pathway in subcutaneous ASCs could be linked to their lower differentiation ability.

## 5. Conclusions

This work demonstrates that the ASCs isolated from subcutaneous and visceral AT share some characteristics, including their cell surface marker profile, cell viability, and cell cycle distribution, but they differ in multiple key aspects, like motility, FA dynamics, secretion of inflammatory cytokines, the expression of stemness related genes, differentiation capability, and primary cilia associated signaling (Figure 7E). These data may be of help for cell-based therapeutic strategies in a wide spectrum of diseases. Heterogeneous results in differentiation and proliferation emphasize the necessity for a standardization of donor selection and matched donor collectives, ASC isolation, characterization, and experimental protocols. Particularly, the donor age and BMI are reported to have great influence on the proliferation and differentiation ability of ASCs. Further investigations are required for studying the interactions of ASCs with their surrounding cells, like immune cells and fibroblasts, and to define their biological functions and molecular mechanisms in vivo via animal models.

## Figures and Tables

**Figure 1 cells-08-01288-f001:**
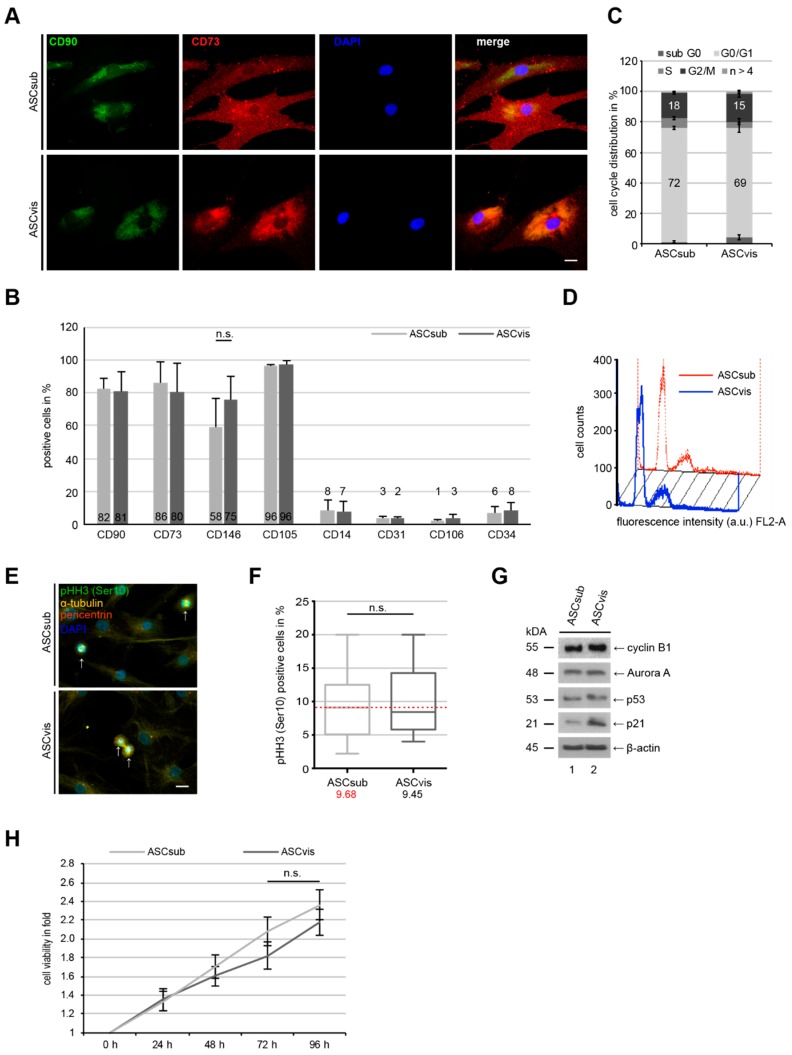
Subcutaneous and visceral adipose-derived mesenchymal stem cells (ASCs) display comparable cell surface marker profiles, cell cycle distribution and cell proliferation. (**A**) Immunofluorescence staining of mesenchymal stem cell surface markers CD90 (green) and CD73 (red), and DNA (DAPI, blue) in subcutaneous ASCs (ASCsub) and visceral ASCs (ASCvis). Scale: 20 μm. (**B**) Flow cytometric analyses of positive cell surface markers CD90, CD73, CD146, and CD105, and negative markers CD14, CD31, CD106, and CD34 for mesenchymal stem cells (MSCs). Values represent the percentages of ASCs expressing the indicated protein. The results from eight independent experiments (donors) are presented as mean ± standard error of the mean (SEM). (**C**,**D**) Cell cycle distribution was analyzed using a FACSCalibur^TM^. Profile examples were shown (**C**). Cell cycle phases of ASCs were presented in percentage and the results were derived from four independent experiments (**D**). (**E**,**F**) ASCs were stained for pHH3 (S10) (green), α-tubulin (yellow), pericentrin (red) and DNA (blue), and representatives are shown (**E**). Scale: 10 μm. pHH3 positive cells were quantified in ASCsub and ASCvis (**F**). The results are from three independent experiments with ASCs from three different donors and presented as median ± min/max whiskers in box plots. n.s. > 0.05. (**G**) Cellular extracts from ASCs were prepared for Western blot analyses with indicated antibodies. β-actin served as loading control. (**H**) ASCs were seeded in 96-well plates for 0, 24, 48, 72, and 96 h. Cell viability was measured via CellTiter-Blue^®^ assay. The results are presented as mean ± SEM and statistically analyzed, showing no significant difference (n.s.).

**Figure 2 cells-08-01288-f002:**
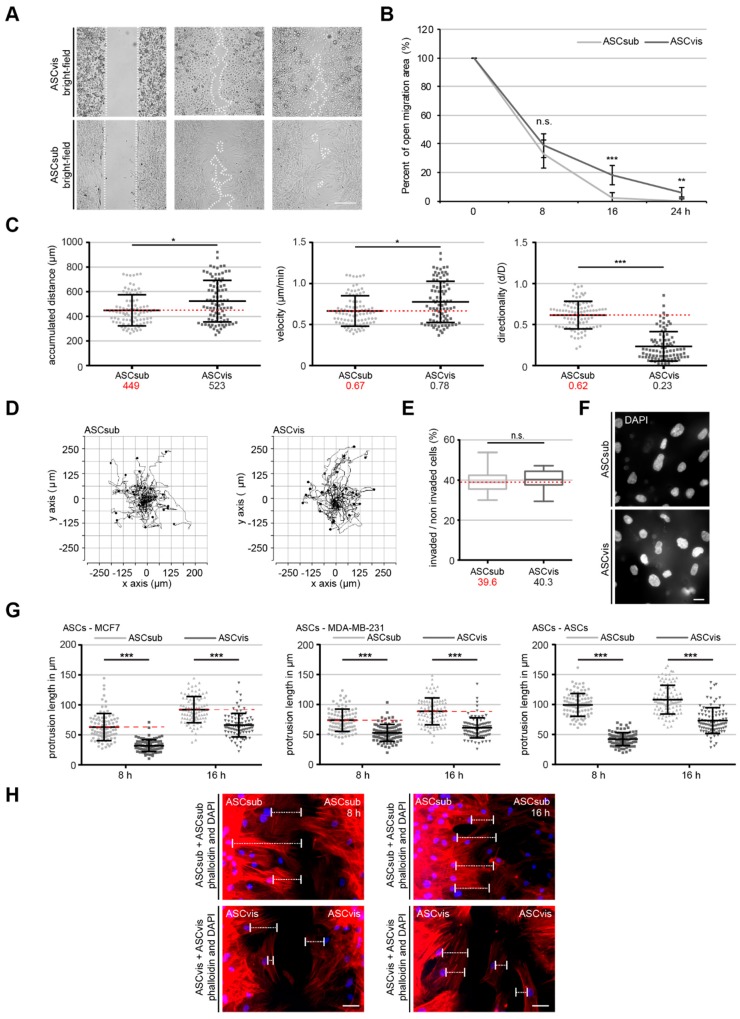
Both ASC subtypes display a comparable motility rate, but subcutaneous ASCs have a significantly higher directed migration capacity. (**A**,**B**) Wound healing/migration assays were performed with subcutaneous and visceral ASCs, and images were taken at indicated time points (0, 8, 16, 24 h) to document the migration front. (**A**) Representatives are shown. White dashed line depicts the migration front. Scale: 300 μm. (**B**) Quantification of the open area between both migration fronts at various time points. The cell-free area at 0 h was assigned as 100%. The results from three independent experiments are presented as mean ± SEM. *** *p* < 0.001. (**C**,**D**) Time-lapse microscopy was performed with subcutaneous or visceral ASCs for up to 12 h. Random motility of these cells was analyzed. (**D**) Representative trajectories of individual cells (*n* = 30) are shown. (**C**) Evaluated accumulated distance (left), velocity (middle), and directionality (right) from three independent experiments are shown as box plots with variations. Unpaired Mann–Whitney *U*-test, * *p* < 0.05, *** *p*< 0.001. (**E**,**F**) Invasion assay. ASCs were seeded into transwells and starved for 12 h. The cells were released into fresh medium for 24 h and fixed for quantification. (**E**) Quantification of invaded cells per field in percent. The results from three independent experiments are presented as mean ± SEM. Student’s t test was performed showing no significant difference (n.s. > 0.05). (**F**) Representatives of invaded ASCs are shown. Scale: 25 μm. (**G**,**H**) Homing assays. ASCs and breast cancer cells were seeded in separated chambers of a culture insert and cultured for 0, 8 and 15 h. (**G**) Evaluation of cell homing distance, the length between the nucleus and the outermost cell protrusion, in subcutaneous and visceral ASCs toward MCF-7 cells (left), MDA-MB-231 cells (middle), and ASC themselves (right). Each experiment was performed in triplicate, and the results are based on three independent experiments and presented as scatter plot showing mean ± SEM. (red dashed line indicates median value of ASCsub). ** *p* < 0.01, *** *p* < 0.001. (**H**) Representatives of ASCs on both migration fronts stained against phalloidin (red) and DAPI (blue) are depicted. White bars indicate cellular protrusion length. Scale: 50 μm.

**Figure 3 cells-08-01288-f003:**
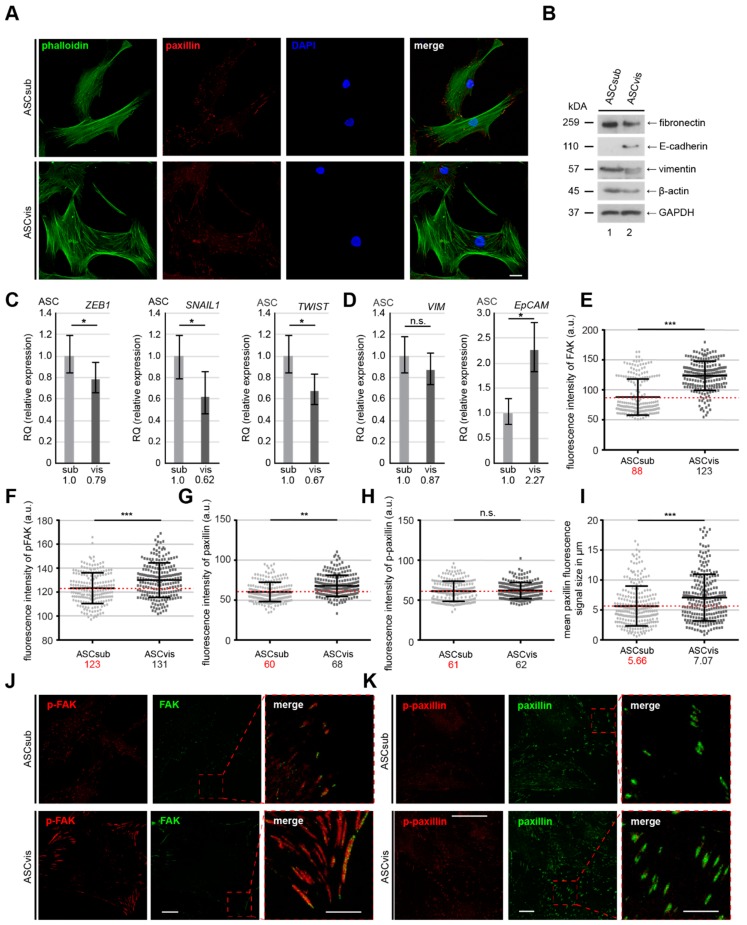
Subcutaneous ASCs show a typical mesenchymal-like phenotype compared to their visceral counterparts. (**A**) Immunofluorescence staining of ASCsub and ASCvis. ASCs were stained for phalloidin (green), paxillin (red) and DNA (blue) to underline their cell morphology. Examples are shown. Scale: 25 μm. (**B**) Cellular extracts from ASCs were prepared for Western blot analyses with antibodies against β-actin, E-cadherin, fibronectin and vimentin. Glyceraldehyde-3-phosphate dehydrogenase (GAPDH) served as loading control. (**C**,**D**) Gene levels of mesenchymal associated transcription factors and cytoskeleton proteins *ZEB1*, *SNAIL, TWIST, VIM,* and *EpCAM* are shown for subcutaneous and visceral ASCs. The results are from three experiments, presented as RQ with minimum and maximum range. RQ, relative quantification of gene expression. Student’s t test, ∗ *p* < 0.05. (**E**–**K**) The focal adhesion composition was analyzed by staining ASCs for focal adhesion kinase (FAK) (green), p-FAK (red), and DNA (DAPI, blue), or for p-paxillin (red), paxillin (green) and DNA (DAPI, blue) for fluorescence microscopy. Quantification of the mean fluorescence intensity of FAK (**E**), p-FAK (**F**)**,** paxillin (**G**) p-paxillin (**H**), and focal adhesion area (**I**) in ASCsub versus ASCvis (at least 200 FAs per staining). The results are based on three independent experiments and presented as scatter plot showing mean ± SEM. Unpaired Mann–Whitney *U*-test, ** *p* < 0.01, *** *p* < 0.001. a.u., arbitrary units. Representatives are depicted (**J**,**K**). Scale: 25 μm.

**Figure 4 cells-08-01288-f004:**
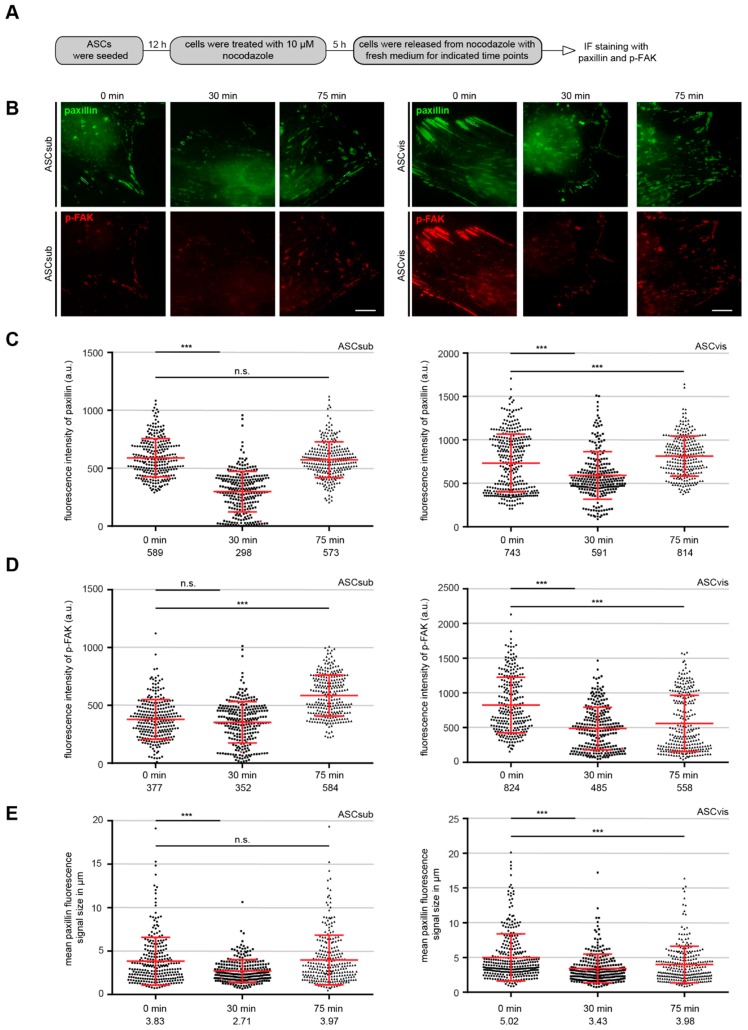
Subcutaneous ASCs dynamically disassemble and reassemble their FAs. (**A**) Schedule of the nocodazole washout assay. (**B**) ASCs were incubated for 5 h with 10 µM nocodazole followed by washout, where the microtubules (MTs) were allowed to regrow for 0, 30 and 75 min. Cells were stained for paxillin (green), p-FAK (red), and DNA (DAPI, blue). Representatives of FA disassembly/reassembly are shown. Scale: 15 µm. (**C**–**E**) Kinetics of FA disassembly during MT regrowth and FA reassembly after MT regrowth. Quantification of the mean fluorescence intensity of paxillin (**C**), p-FAK (**D**), and FA size (**E**) (270 FA per condition) is depicted. The results are based on three independent experiments and presented as scatter plots showing mean ± SEM. Unpaired Mann–Whitney U-test, ** *p* < 0.01, *** *p* < 0.001. a.u., arbitrary units.

**Figure 5 cells-08-01288-f005:**
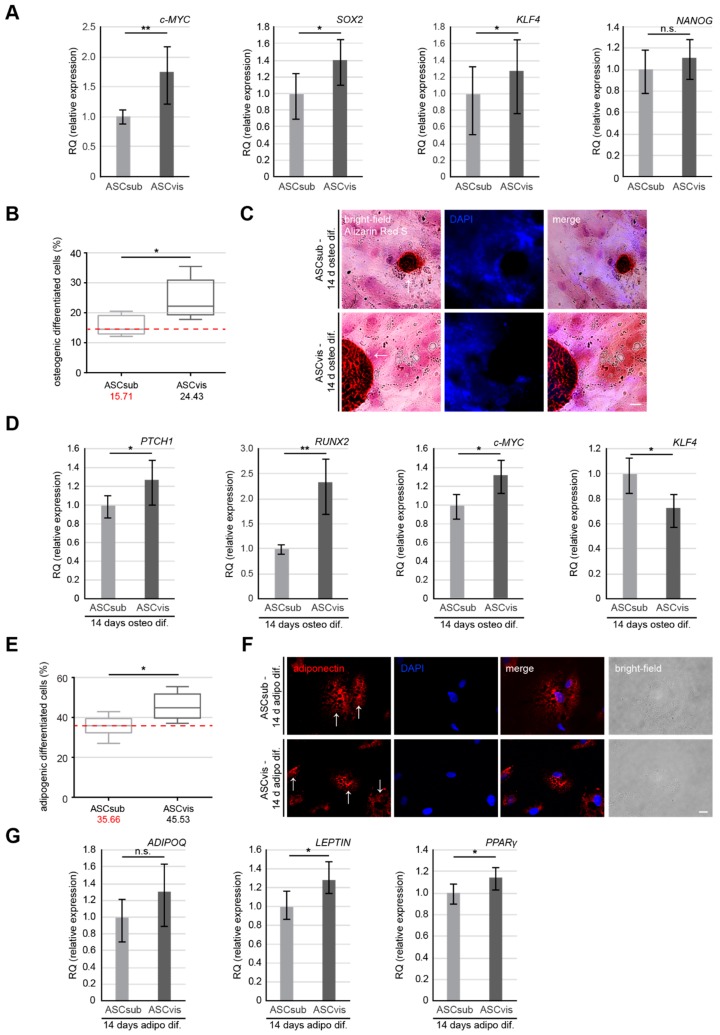
Visceral ASCs are superior in osteogenic and adipogenic differentiation compared to subcutaneous ASCs. (**A**) Gene levels of stemness/self-renewal associated genes *c-MYC*, *SOX2*, *KLF4,* and *NANOG* are shown for subcutaneous and visceral ASCs. The results are from three experiments, presented as RQ with minimum and maximum range. Student’s t test, ∗ *p* < 0.05, ** *p* < 0.01. (**B**–**D**) ASCsub and ASCvis cells were induced into osteogenic differentiation for up to 14 days. The percentage of differentiated ASCs was evaluated by Alizarin Red S staining. (**B**) The quantification data are presented as median ± min/max whiskers (red dashed line indicates median value of ASCsub, *n* = 300 cells for each condition, pooled from three experiments). Student’s t test, ∗*p* < 0.05. (**C**) Example images for Alizarin Red S staining are shown. Scale: 20 μm. (**D**) Expression levels of differentiation related genes *PTCH1* (1^st^ graph), *RUNX2* (2^nd^ graph), *KLF4* (3^rd^ graph), and *c-MYC* (4^th^ graph) in differentiated subcutaneous and visceral ASCs. The results are from three experiments and presented as RQ with minimum and maximum range. Student’s t test, ∗ *p* < 0.05, ** *p* < 0.01. (**E**–**G**) Analyses of cells with lipid vacuoles after 14 days of adipogenic differentiation. (**E**) The quantification shows the percentage of cells differentiated into adipogenic-like cells. Results are presented as median ± min/max whiskers in visceral ASCs (*n* = 200 cells for each condition, pooled from three experiments) and the red dashed line illustrates the median value of ASCsub. Student’s t test, ∗ *p* < 0.05. (**F**) Representative images of cells displaying lipid vacuoles stained for adiponectin (red) and DNA (DAPI, blue). (**G**) Gene levels of *ADIPOQ, LEPTIN,* and *PPARγ* after adipogenic differentiation are shown for subcutaneous and visceral ASCs. The results are from three experiments, presented as RQ with minimum and maximum range. Student’s t test, ∗ *p* < 0.05.

**Figure 6 cells-08-01288-f006:**
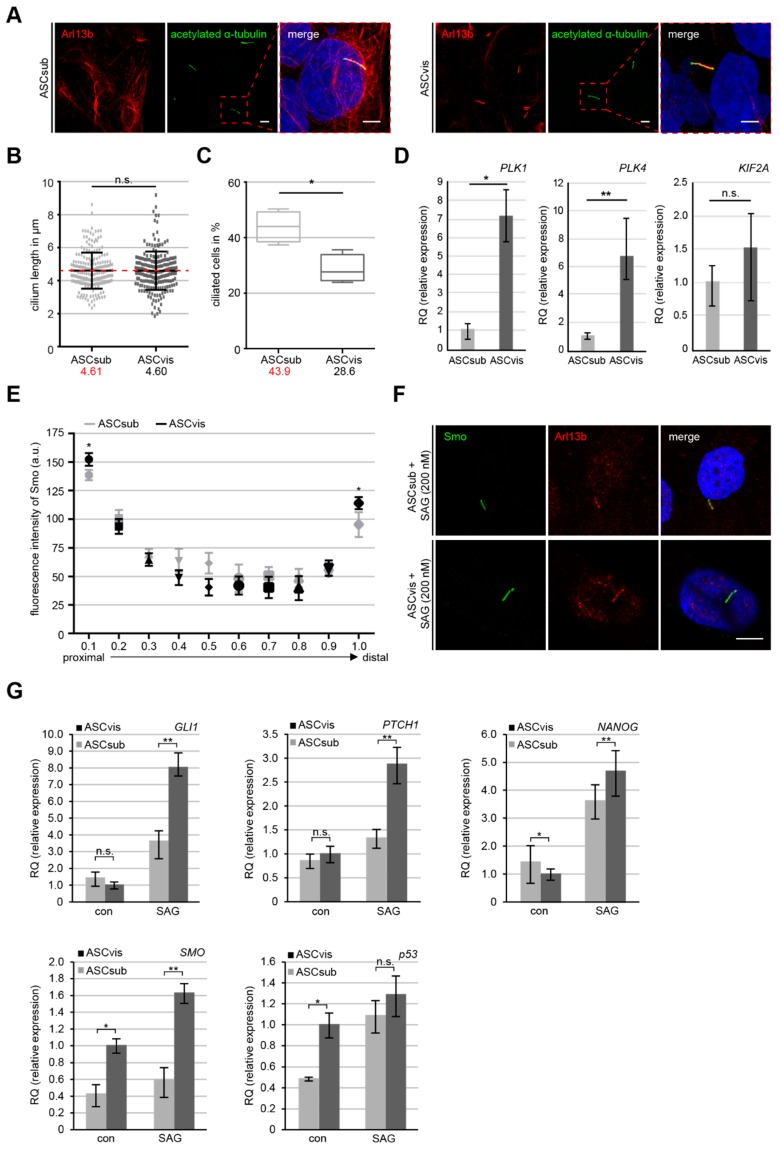
Visceral ASCs display high deciliation gene levels and enhanced activation of the sonic hedgehog (Hh) signaling pathway. (**A**) Primary cilia of ASCsub and ASCvis were stained for acetylated α-tubulin and Arl13b. Representatives are shown. Scale: 10 μm. Regions outlined in boxes are shown in a higher magnification. Inset scale: 10 µm. (**B**) The cilium length was evaluated. The results are based on six experiments using ASCs from six donors (*n* = 180 cilia for each group). (**C**) Ciliated ASCs were evaluated and the results are presented as mean ± SEM (*n* = 600 cells, pooled from six experiments). Unpaired Mann–Whitney *U*-test, * *p* < 0.05. (**D**) The gene levels of deciliation regulators *PLK1*, *PLK4,* and *KIF2A*. The data are based on three experiments and presented as RQ with minimum and maximum range. Student’s t test, ∗ *p* < 0.05, ** *p* < 0.01. (**E**–**G**) Fluorescence intensities and expression levels of important genes related to the Hh pathway are shown for ASCs treated with SAG for 24 h. (**E**) Each point of the curve represents the mean fluorescence intensity (mean ± SEM) based on three experiments (*n* = 30 cilia). Unpaired Mann–Whitney *U*-test, * *p* < 0.05. (**F**) Representatives are shown for measurements of primary cilium staining of acetylated α-tubulin, Arl13b and Smoothened (Smo). Scale: 3 μm. (G) The gene levels of *GLI1*, *PTCH1*, *NANOG*, *SMO,* and *TP53* are shown for ASCs treated or non-treated with 200 nM SAG for 24 h. The results are from three experiments, merged as biological group, and presented as mean ± SEM. Student’s t test, ∗ *p* < 0.05, ** *p* < 0.01.

**Figure 7 cells-08-01288-f007:**
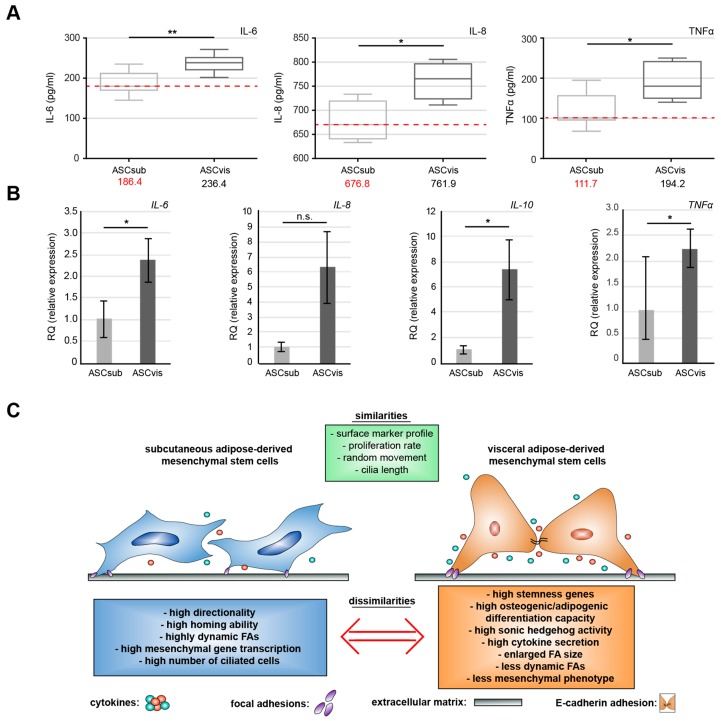
Visceral ASCs secrete more pro-inflammatory cytokines. (**A**) The supernatants of subcutaneous and visceral ASCs in the third passage were collected after 72 h culture and used for evaluation of IL-6 (left), IL-8 (middle) and TNF-α (right) by enzyme-linked immunosorbent assay (ELISA). The results are from four experiments and presented as median ± min/max whiskers in box plots. Student’s t test, ^∗^
*p* < 0.05, ** *p* < 0.01. (**B**) The gene levels of *IL-6*, *IL-8*, *IL-10,* and *TNFα*. The data are based on three experiments and presented as RQ with minimum and maximum range. Student’s t test, ^∗^
*p* < 0.05. (**C**) Schematic illustration of the proposed similarities and dissimilarities between both ASC subtypes. The key dissimilarities between subcutaneous and visceral ASCs are their migration mode, differentiation capacity, and cytokine secretion, which affect a variety of different pathways, like the Hh signaling on the primary cilium.

**Table 1 cells-08-01288-t001:** Clinical information of 16 participants.

	Age	Gestational Age (weeks)	Body Mass Index (BMI)	Birth Weight (g)
mean value	31.6 ± 4.6	37.7 ± 2.8	24.1 ± 2.9	2964 ± 581

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
