# Peer review of "Subcutaneous and Visceral Adipose-Derived Mesenchymal Stem Cells: Commonality and Diversity"

_cells, 2019, doi:10.3390/cells8101288_

Round 1
Reviewer 1 Report
In the current study, the authors have demonstrated commonality and diversity between Subcutaneous and visceral adipose-derived mesenchymal stem cells. Overall, it is a well presented manuscript which is of relevance to the audience of journal. However, I have following concerns related to this study:
Since visceral adipose tissues surrounds organs, I could not understand the rationale of its clinical use, as it could add risk to nearby organs while isolating V-ASC. Though many previous studies have demonstrated common biological properties shared by S-ASC and V-ASC, the current study has further detailed provided cell cycle distribution, pHH3 (mitotic marker), and mitotic proteins cyclin B1 and Aurora A, the results of which are very much informative. Previous studies have already revealed enhanced adipogenic potential of S-ASC (doi: 10.1111/jcmm.13138), which is contradictory to the current study. Authors should consider to include such outcomes in their discussion section. Conclusion section should not have references and the overall study could be concluded within 3-4 lines. There are many minor spelling errors in the manuscript such as The word “secrete” has been misspelled in the overall manuscript. line no. 517: it should be “showed”.Author Response
Dear colleague,
Thank you for your kind revision of this paper and the constructive suggestion concerning the contradictory differentiation results. We added this topic in the manuscript under the following lines:
382-385, 555-560 and 580-582.
In my opinion, two of the most crucial requirements in the field of stem cell- and especially ASC research are first the usage of samples from patients matched as good as possible and second to standardize the isolation and culture methods of these cells. In the very nice paper you mentioned in your comment, the researcher used rats and human patients with an inhomogeneous age range (23-49) and without stating their BMI, which might explain the divergent data.
Additionally, we removed the citations in the conclusion, also compressed this part and corrected thoroughly the spelling errors in the whole manuscript.
Reviewer 2 Report
Dear colleagues!
It is comprehensive comparative study of biological properties of MSC isolated from 2 sources - subcutaneous fat and visceral fat of matched donors.
The study contains numerous important findings which go beyond descriptive level and highlight intrinsic differences that arise in these 2 cell types including transcription factor profile and signaling pathways that are differentially activated in these cell types.
Overall, it has impact for an important problem of tissue-specificity which is often neglected when we talk of cell-based regenerative therapies.
I congratulate the authors on this thoroughly-designed set of original data and find it of importance for the field and the Journal reputation.
Author Response
Dear colleague,
Thank you for your very kind revision!
We performed some minor changes concerning the discussion and checked the spelling of the whole manuscript.